# Facile Synthesis of Black Phosphorus Nanosheet@NaReF_4_ Nanocomposites for Potential Bioimaging

**DOI:** 10.3390/nano12193383

**Published:** 2022-09-27

**Authors:** Dongya Wang, Jingcan Qin, Chuan Zhang, Yuehua Li

**Affiliations:** 1Department of Radiology, Shanghai Jiao Tong University Affiliated Sixth People’s Hospital, Shanghai Jiao Tong University School of Medicine, 600 Yi Shan Road, Shanghai 200233, China; 2School of Chemistry and Chemical Engineering, Frontiers Science Center for Transformative Molecules, State Key Laboratory of Metal Matrix Composites, Shanghai Jiao Tong University, 800 Dongchuan Road, Shanghai 200240, China

**Keywords:** black phosphorus, rare-earth nanoparticle, nanocomposites, magnetic resonance imaging, photoacoustic imaging

## Abstract

Black phosphorus nanomaterials (BPN) have been well developed in tumor therapy. However, lack of diagnostic function limits the development of BPN in biomedicine. Lanthanide-doped nanoparticles are considered as versatile materials for fluorescence or magnetic resonance imaging. Integration of BPN with lanthanide-doped nanoparticles was rarely reported owing to the complex synthesis processes and poor modification effect. Herein, we report a simple and general method for synthesizing BPN@NaReF_4_ (Re: Gd or Y, Yb, Er) nanocomposite. TEM and XRD characterization confirm efficient combination of BPN and NaGdF_4_ or NaYF_4_:Yb,Er (18.2 mol %) after one-step mixing. The FTIR and XPS spectra were used to prove the generation of PO_4_^3^^−^-Gd and P-Gd coordination bonds and clarify ligand exchange mechanism. The anchored nanoparticles on BPN were stable and become hydrophilic. The prepared BPN@NaGdF_4_ exhibit the signals of photoacoustic and magnetic resonance imaging. The obtained BPN@NaYF_4_:Yb,Er (18.2 mol %) have the potential in fluorescence bioimaging. We believe that this work will expand the applications of BPN in diagnosis and therapy together.

## 1. Introduction

Black phosphorus, a two-dimensional semiconductor, has attracted much attention from researchers in recent years due to its unique layer-dependent properties [1]. The black phosphorus nanosheet (BPN), exfoliated from its bulk counterpart, exhibits outstanding optical, electronic, thermal, and catalytic performances, and, thus, may be widely applied in optoelectronics [2], energy conversion [3], and biomedicine [4]. Nevertheless, to meet the diversified demand in varied fields, the functionality of black phosphorus nanomaterials should be further improved, especially when used in biomedicine.

With the advantages of high specific area, good biocompatibility, and photothermal/photodynamic properties, BPN was well studied for tumor treatment and showed great therapy performance [5,6]; however, lack of diagnostic function limits the development of BPN in clinical use. To improve it, imaging medium, such as Cy7 dyes [7] or iron oxide [8], was loaded with BPN, realizing bioimaging and therapy together. Sadly, the binding force between BPN and contrast medium is an electrostatic interaction which will cause uncontrollable aggregation of nanomaterial and medium leakage [9,10]. Thus, continual effort in developing effective strategies should be made to endow BPN with bioimaging functions, such as fluorescence imaging or magnetic resonance imaging, which is very common and powerful in clinics [11].

Lanthanide-doped nanoparticles, also called rare-earth nanoparticles (NaReF_4_), have been developed as an important class of functional nanomaterials for many years. Benefiting from the inner-layer electron transition, lanthanide-doped nanoparticles commonly exhibit light emission with large anti-stokes shift, long luminescence lifetimes, and excellent photostability [12,13]. Moreover, Gd^3+^ based complex and nanoparticle yield magnetic resonance signals [14]. Thus, lanthanide-doped nanoparticles were widely used for disease diagnosis, including fluorescence imaging and magnetic resonance imaging [15,16,17]. In this regard, integration of lanthanide-doped nanoparticles with BPN to form 0D-2D heterostructure will be an upward trend, realizing integrated diagnostic and therapy. Unfortunately, to the best of our knowledge, the rare-earth nanoparticles were married with BPN just via electrostatic attraction which often suffers inefficient modification and intricate procedures including charged polymer coating or surface modification [18,19]. A simple method of combining rare-earth nanoparticles with BPN is desirable and worthwhile for nanotechnology development.

In this work, we synthesized BPN@NaReF_4_ (Re: Gd or Y, Yb, Er) heterostructure by a simple process. As the schematics illustration show in Figure 1, the oleic acid-capped NaReF_4_ in cyclohexane was directly mixed with BPN in water. After vortexing overnight, the NaReF_4_ in the oil phase was transferred to the aqueous phase and integrated with BPN, forming NaReF_4_ coordinated BPN nanocomposites. The prepared BPN@NaGdF_4_ nanocomposites exhibit photoacoustic and magnetic resonance signals and the BPN@NaYF_4_:Yb,Er (18.2 mol %) demonstrate fluorescence emission, showing the potential application in bioimaging.

## 2. Materials and Methods

### 2.1. Materials

Bulk black phosphorus crystals were synthesized via a chemical vapor transport method according to our previously reported procedures [20]. N-Methyl-2-pyrrolidone (NMP, 98%), cyclohexane (99.5%), methanol (99.8%), and ethanol (AR) were obtained from Aladdin Biochemical Technology Co., Ltd. (Shanghai, China). Hydrochloric acid (~36%) and sodium hydroxide (>98%) were purchased from Sinopharm Chemical Reagent Co., Ltd. Yttrium(III) acetate hydrate (99.9%), ytterbium(III) acetate hydrate (99.9%), erbium(III) acetate hydrate (99.9%), gadolinium(III) acetate hydrate (99.9%), and oleic acid (OA, 90%) were purchased from Alfa Aesar (Haverhill, MA, USA). Ammonium fluoride (>98%) and 1-Octadecene (90%) was purchased from Sigma-Aldrich (St. Louis, MO, USA). All chemicals were used as received without further purification.

### 2.2. Characterization

Low-resolution TEM images were taken from a Talos L120C G2 (USA) instrument operated at 120 kV. Powder X-ray diffraction (XRD) patterns were recorded using a Rigaku Mini Flex 600 with Cu Kα radiation (λ = 1.5406 Å). UV-Vis spectra were recorded using a UV-1800 instrument (Shimadzu, Kyoto, Japan). Fluorescence spectra were obtained on an FLS1000 fluorescence spectrometer (Edinburgh Instruments, Livingston, UK). Scanning electronmicroscopy (SEM) images were carried out in a JSM-7800F (JEOL) scanning electron microscope. Raman spectra were obtained using a Renishaw inVia Qontor confocal microscope with a 532 nm laser. FTIR spectra were obtained on a Nicolet iN10 MX instrument (Thermo Scientific, Waltham, MA, USA). X-ray photoelectron spectroscopy (XPS) was performed on an AXIS Ultra DLD (Shimadzu, Kyoto, Japan). Inductively coupled plasma mass spectrometry (ICP-MS, iCAP Q, Thermal Fisher, Waltham, MA, USA) was used to determine the concentration of black phosphorus and gadolinium. Photoacoustic signal was obtained on a VEVO LAZR-X (Fujifilm VisualSonics, Toronto, Canada). The magnetic resonance images and T_1_ relaxation rate were acquired on the 0.5 T MesoMR23-060H-I (Niumai Electronic Technology, Suzhou, China).

### 2.3. Synthesis of Black Phosphorus Nanosheets

Bulk BP (200 mg) was ground with NMP and then dispersed in NMP (100 mL). The mixture was first tip-sonicated (working 2 s, interval 2 s, power 650 W) for 3 h in an ice bath, followed by ice bath sonication for 12 h at 300 W. The resulting suspension was centrifuged at 7000 rpm for 20 min, and the collected supernatant was then centrifuged at 14,000 rpm for 20 min. The precipitate was washed with DI water three times and dispersed in the aqueous solution for further use. The concentration of BPN dispersion was determined by ICP-MS.

### 2.4. Synthesis of NaGdF_4_ Nanoparticles

Oleic acid capped NaGdF_4_ (NaGdF_4_-OA) nanoparticles were prepared using the thermal coprecipitation method according to the literatures [21]. Briefly, oleic acid (4 mL), 1-octadecene (6 mL), and an aqueous solution (2 mL) containing Gd(CH_3_COO)_3_ (0.4 mmol) were added to a 50 mL two-neck round-bottom flask. The mixture was stirred at 150 °C for 1 h to remove the water. After the mixture was cooled to room temperature, a methanol solution (5 mL) containing NaOH (1 mmol) and NH_4_F (1.2 mmol) was added to the flask. The mixture was then stirred at 50 °C for 2 h, followed by maintaining the temperature at 100 °C for another 30 min. Subsequently, the mixture was heated to 280 °C and kept for 2 h under nitrogen atmosphere. After cooling down to room temperature, the oleic acid capped NaGdF_4_ nanoparticles were collected by centrifugation and washed three times with ethanol. The nanoparticles were dispersed in cyclohexane for further use.

### 2.5. Synthesis of NaYF_4_:Yb,Er (18.2 mol %) Nanoparticles

Oleic acid capped NaYF_4_:Yb,Er (18.2 mol %) nanoparticles were also prepared by the thermal coprecipitation method according to our previous work [22]. The Y(CH_3_COO)_3_ (0.32 mmol), Yb(CH_3_COO)_3_ (0.072 mmol) and Er(CH_3_COO)_3_ (0.008 mmol) aqueous solution was mixed with oleic acid (3 mL) and 1-Octadecene (7 mL). After stirring at 150 °C for 1 h, the mixture was cooled to room temperature, and then NaOH (1 mmol) and NH_4_F (1.2 mmol) dissolved in methanol was added followed by stirring at 50 °C for 2 h. Next, the mixture was heated to 100 °C and kept for another 30 min. After purging with nitrogen, the reaction mixture was heated to 290 °C and kept for 1.5 h before cooling down to room temperature. The resulting nanoparticles were collected by centrifugation washed with ethanol. The oleic acid capped NaYF_4_:Yb,Er (18.2 mol %) nanoparticles were finally obtained and dispersed in cyclohexane.

### 2.6. Synthesis of Ligand-Free Nanoparticles

Ligand-free rare-earth nanoparticles were prepared according to a procedure in the literature, with slight modification [23]. The oleic acid capped nanoparticles were dispersed in ethanol (1 mL), and after adding hydrochloric acid (1 mL, 1 M), the mixture was ultrasonicated for 1 min. The ligand-free nanoparticles were collected by centrifugation at 14,000 rpm for 20 min and washed with ethanol three times. The ligand-free rare-earth nanoparticles were redispersed in water for the MRI test.

### 2.7. Synthesis of BPN@NaReF_4_ Nanocomposites

The oleic acid capped NaGdF_4_ or NaYF_4_:Yb,Er (18.2 mol %) in cyclohexane (2 mL, 50~500 μg mL^−1^) was directly mixed with BPN aqueous solution (2 mL, 100 μg mL^−1^). After vortex overnight, the nanoparticles in cyclohexane were transferred to water and integrated with BPN in the meantime. The BPN@NaGdF_4_ or BPN@NaYF_4_:Yb,Er (18.2 mol %) nanocomposites were collected from the aqueous phase by centrifugation.

## 3. Results

### 3.1. Characterization of BPN@NaGdF_4_ Nanocomposites

NaGdF_4_ nanocrystals were used as the representative lanthanide-doped nanoparticles to examine the integration between BPN and NaReF_4_. First, the BPN and NaGdF_4_ were synthesized separately and characterized with TEM, XRD, absorption spectra, and Raman spectra.

As the XRD pattern shows in Appendix A, the exfoliated black phosphorus nanosheets still have the crystal structure. The lateral size of obtained BPN was 100–200 nm (Figure 2A). The optical bandgap derived from the BPN absorption spectrum was ∼2.18 eV (Figure 3A,B), indicating the ultrathin thickness of the obtained BPN [24]. The Raman spectrum (Figure 3C) shows three characteristic peaks at 361, 438, and 466 cm^−1^, assigned to the modes of A_g_^1^, B_2g_, and A_g_^2^, respectively, demonstrating the puckered orthorhombic lattice structure of the exfoliated BPN [25].

The synthesized NaGdF_4_ nanoparticles, capped by oleic acid, have high uniformity in size (average size of ∼8.3 nm) as shown in Figure 2B and Appendix A. The XRD pattern proves that the obtained NaGdF_4_ nanoparticles are composed of pure hexagonal NaGdF_4_ crystals (Appendix A).

After the integration of NaGdF_4_ and BPN, the product was collected from the aqueous phase. As Appendix A shows, the amount of NaGdF_4_ nanoparticles was decreased which indicates the possible combination of BPN and NaGdF_4_. From the TEM image in Figure 2C, we can see that the nanoparticles were distributed on the surface of BPN and no free nanoparticles were found. XRD pattern in Figure 2D shows combined diffraction peaks of BPN and NaGdF_4_ nanocrystal. After NaGdF_4_ coating, the Raman peaks of BPN experience a slight red shift, indicating that the NaGdF_4_ nanoparticles were successfully loaded on the BPN [26]. To investigate the loading capacity of BPN for NaGdF_4_ nanoparticles, different amounts of NaGdF_4_ nanoparticles were added in the oil phase. After the reaction, the collected products were characterized with TEM (Appendix A). It is obvious that the numbers of loaded NaGdF_4_ nanoparticles are increased with the elevated feeding ratios of NaGdF_4_ to BPN and keep constant when the ratio is higher than 2. Above all, these results confirm the efficient integration of NaGdF_4_ with BPN via this simple experimental method.

It should be noted that the optical bandgap of BPN@NaGdF_4_ calculated from the absorption spectrum is generally consistent with BPN (Figure 3A,B). Due to the layer-dependent bandgap, the optical bandgap can reflect the thickness and aggregation degree [27]. Thus, it is concluded that the rigid NaGdF_4_ nanoparticles can prevent aggregation of BPN after loading them.

### 3.2. Mechanism of BPN@NaGdF_4_ Preparation

To investigate the reason why NaGdF_4_ can be bound with BPN, the FTIR and XPS spectra were acquired. For the FTIR spectrum of BPN@NaGdF_4_ nanocomposites (Figure 4A), the peaks at 2920, 2854, 1554, and 1454 cm^−1^ disappeared, unlike with OA capped NaGdF_4_, which confirms removal of the OA [22]. According to the literature, the BPN is easily oxidized and PO_4_^3−^ can be generated on the surface of BPN [28,29]. From the FTIR spectrum of BPN, we can recognize the peaks located in the 1115–975 cm^−1^ range which are from PO_4_^3−^ [30,31]. The XPS spectra (Figure 4B) also show the binding energy of PO_x_ especially for the BPN@NaGdF_4_ sample. Thus, the PO_4_^3−^ were indeed existed on the as-prepared BPN and BPN@NaGdF_4_ nanocomposites.

It is well known that the PO_4_^3–^ can easily replace the CO_2_^−^, as the PO_4_^3−^ has a stronger coordination ability for rare earth than carboxyl [32,33]. In this case, we believe that the oleic acid with CO_2_^−^ was displaced by PO_4_^3−^ on the BPN, and PO_4_^3–^-Gd coordination was formed. Thus, the NaGdF_4_ nanoparticles can be integrated with BPN.

In addition to the coordination of PO_4_^3−^ for Gd element, the P-Gd coordination [34] has also been studied. As shown in Figure 4B–D, after the combination of BPN and NaGdF_4_, the XPS peaks of P 2p and Gd 4d are shifted to higher binding energy while the Gd 3d XPS peaks exhibit no change, indicating strong interaction between P and Gd atoms. By XPS-peak-differentiation-imitating analysis for P 2p and Gd 4d XPS spectra [35,36], there are two new emerged peaks that can all be assigned to P-Gd coordination.

Generally, the coordination bond is stable. In this case, we examined whether the anchored NaGdF_4_ on BPN can be dropped or not. The prepared BPN@NaGdF_4_ nanocomposites in water were stirred vigorously overnight and TEM was conducted. In Appendix A, almost no free NaGdF_4_ nanoparticles can be observed. Thus, the anchored NaGdF_4_ nanoparticles on BPN by this method are stable.

Above all, we concluded that the PO_4_^3−^ groups on BPN replace the CO_2_^−^ contained oleic acids on the surface of NaGdF_4_ and PO_4_^3−^-Gd as well as P-Gd coordination bonds are formed, realizing firm integration of NaGdF_4_ and BPN.

To prove the generality of such synthesis method, NaYF_4_:Yb,Er (18.2 mol %) nanoparticles with an average size of 22 nm were prepared to modify BPN (Figure 5A,B). After the modification process, the amount of NaYF_4_:Yb,Er (18.2 mol %) nanoparticles in upper cyclohexane was reduced (Appendix A) and the TEM image of the product shows that the NaYF_4_:Yb,Er (18.2 mol %) nanoparticles were effectively anchored on BPN (Figure 5C). Consequently, we believe that the described experimental method in this work can be used to integrate other lanthanide-doped nanoparticles with BPN.

### 3.3. Bioimaging Properties of BPN@NaReF_4_ Nanocomposites

The black phosphorus nanomaterials have been reported exhibit photoacoustic property owing to the excellent photothermal conversion and thermal stability [37,38,39]. We then examine the photoacoustic signal of BPN and BPN@NaGdF_4_ nanocomposites. The photoacoustic signal of BPN was strongest when the excitation wavelength was 710 nm (Appendix A). The photoacoustic intensities all show good linear positive correlations with the concentrations of BPN (Figure 6A,B). The BPN@NaGdF_4_ nanocomposites with 200 μg mL^−1^ BPN display obvious photoacoustic imaging effect.

Gd-based chelates (Gd-DTPA, Gd-DOTA, etc.), as a class of T_1_ contrast agent, have been widely used in routine clinical magnetic resonance imaging (MRI) diagnosis [40,41]. NaGdF_4_ nanoparticles have also been well studied as contrast agent and show enhanced MRI effect [15,16]. In this case, the relaxation rate of NaGdF_4_ nanoparticles and BPN@NaGdF_4_ were evaluated. As shown in Figure 6C,D and Appendix A, the NaGdF_4_ nanoparticles after BPN loading still exhibit comparable MRI signal compared with ligand-free NaGdF_4_.

The NaYF_4_:Yb,Er (18.2 mol %) nanoparticles are considered to be the more efficient NIR-to-visible upconverting nanomaterials, showing much potential in bioimaging application. As Figure 7 shows, when excited by a 980 nm laser, NaYF_4_:Yb,Er (18.2 mol %) nanoparticles have three emission peaks. After loading with BPN, the emission peaks in 507–575 nm range are quenched more severely owing to stronger absorption of BPN for shorter wavelength light (Figure 2A). The emerged emission peaks at 647 nm and 697 nm may be the results of light scattering or fluorescence emission of BPN. We speculate that there could be energy transfer between BPN and NaYF_4_:Yb,Er (18.2 mol %) and it will be further investigated in the future. In a word, the BPN@NaYF_4_:Yb,Er (18.2 mol %) nanocomposites still exhibit satisfied fluorescence emission for bioimaging application.

## 4. Conclusions

In this work, the BPN@NaReF_4_ (Re: Gd or Y, Yb, Er) nanocomposites were successfully synthesized via simple mixing. The formation mechanism of BPN@NaGdF_4_ is that the oleic acid on the surface of NaGdF_4_ can be substituted with the PO_4_^3^^−^ on the surface of BPN. Coordination bonds (PO_4_^3^^−^-Gd and P-Gd) are generated between Gd and PO_4_^3^^−^ or P, so that the modified NaGdF_4_ on BPN are stable. The prepared BPN@NaGdF_4_ nanocomposites have the properties of photoacoustic and magnetic resonance imaging. The BPN@NaYF_4_:Yb,Er (18.2 mol %) nanocomposites in aqueous still show satisfied fluorescence emission and can be used for fluorescence bioimaging in future.

## Figures and Tables

**Figure 1 nanomaterials-12-03383-f001:**
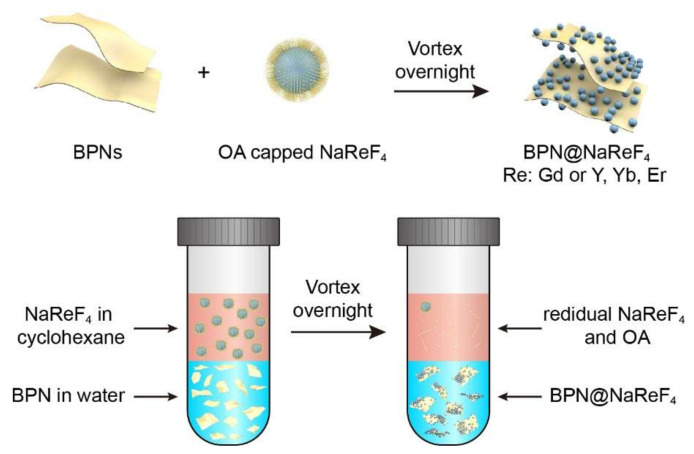
Schematic illustrations of BPN@NaReF_4_ synthesis and corresponding experimental process.

**Figure 2 nanomaterials-12-03383-f002:**
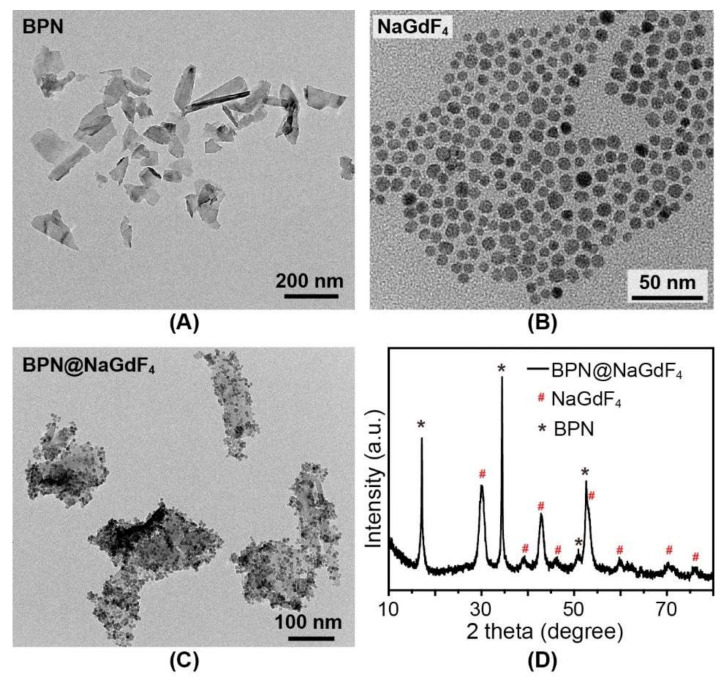
The TEM images of BPN (**A**), NaGdF_4_ (**B**), and BPN@NaGdF_4_ (**C**); (**D**) XRD pattern of the as-synthesized BPN@NaGdF_4_ nanocomposites.

**Figure 3 nanomaterials-12-03383-f003:**
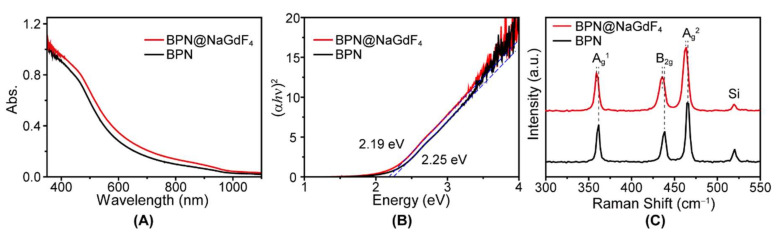
(**A**) The absorption spectra of BPN and BPN@NaGdF_4_; (**B**) direct Tauc plots used to determine the optical bandgap. *α* is the absorbance and *hν* is the photon energy of the incident light; and (**C**) Raman spectra of BPN and BPN@NaGdF_4_.

**Figure 4 nanomaterials-12-03383-f004:**
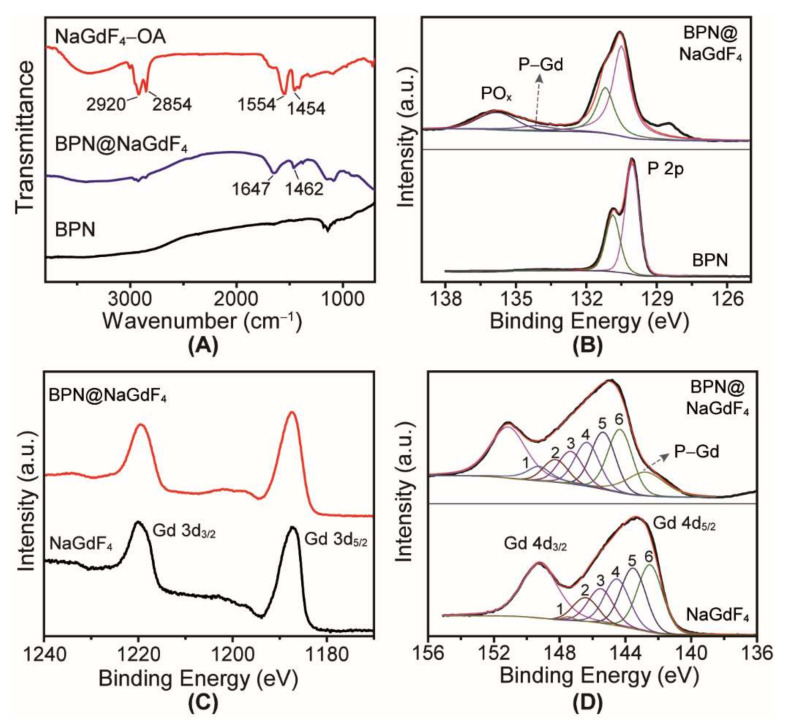
(**A**) FTIR spectra of BPN, NaGdF_4_, and BPN@NaGdF_4_ nanocomposites; (**B**) high-resolution P 2p XPS spectra of BPN and BPN@NaGdF_4_ nanocomposites; high-resolution Gd 3d (**C**) and Gd 4d (**D**) XPS spectra of NaGdF_4_ and BPN@NaGdF_4_ nanocomposites.

**Figure 5 nanomaterials-12-03383-f005:**
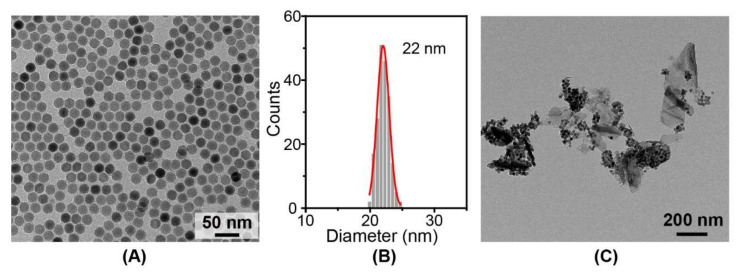
TEM image of OA capped NaYF_4_:Yb,Er (18.2 mol %) nanoparticles (**A**) and the corresponding size distribution (B); (**C**) TEM image of BPN@NaYF_4_:Yb,Er (18.2 mol %) nanocomposites.

**Figure 6 nanomaterials-12-03383-f006:**
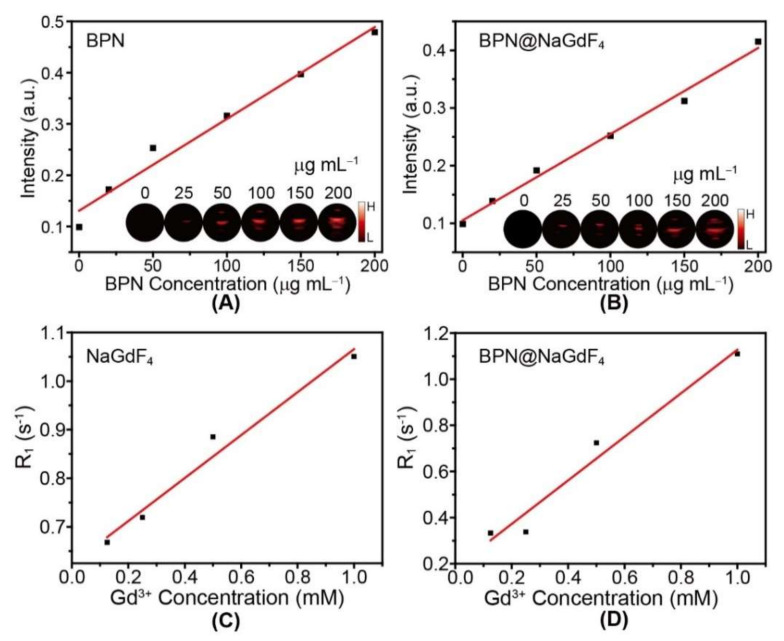
Photoacoustic intensity of BPN (**A**) and BPN@NaGdF_4_ (**B**) with different concentrations of BPN and their linear fit. Inset: corresponding photoacoustic imaging; Relaxation rate R_1_ (1/T_1_) of ligand-free NaGdF_4_ (**C**) and BPN@NaGdF_4_ (**D**) versus Gd^3+^ concentration.

**Figure 7 nanomaterials-12-03383-f007:**
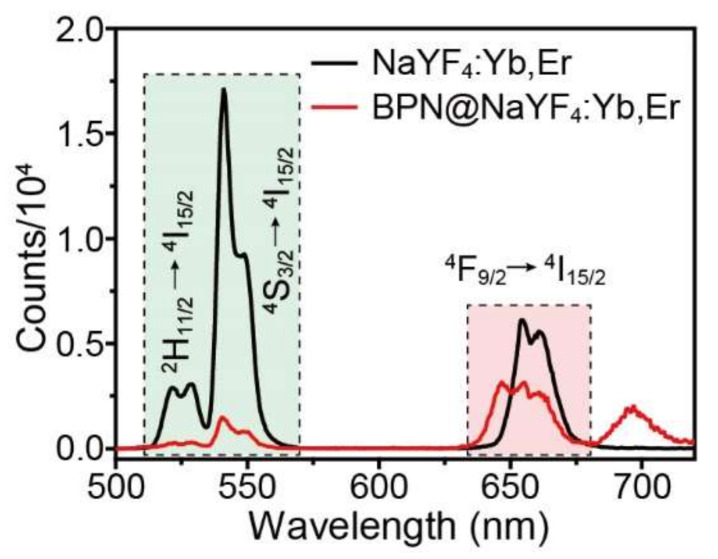
Room-temperature upconversion emission spectra of the oleic acid capped NaYF_4_:Yb,Er (18.2 mol %) nanoparticles dispersed in cyclohexane and BPN@NaYF_4_:Yb,Er (18.2 mol %) in water.

## Data Availability

The data presented in this study are available on request from the first author.

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
