# Peer review of "Facile Synthesis of Black Phosphorus Nanosheet@NaReF4 Nanocomposites for Potential Bioimaging"

_nanomaterials, 2022, doi:10.3390/nano12193383_

Round 1
Reviewer 1 Report
The paper entitled “Facile Synthesis of Black Phosphorus Nanosheet@NaReF4 Nanocomposite for Potential Bioimaging” was interesting and well written by the authors. All the necessary characterizations were done and the discussion has been supported by the experimental data. However, the manuscript was very confusing because the authors presented some properties of BPN@NaGdF4 and some properties of BPN@NaYF4 nanocomposite. For example, photoacoustic property for BPN@NaGdF4 nanocomposite and luminescence property for BPN@NaYF4 nanocomposite. It would be better if the authors present the comparison between the photoacoustic and luminescence properties of these two nanocomposites. Also, the authors need to present SEM images of the BPN nanosheet, NaGdF4 nanoparticles, and the BPN@NaGdF4 nanocomposite.
Author Response
Point 1: The paper entitled “Facile Synthesis of Black Phosphorus Nanosheet@NaReF4 Nanocomposite for Potential Bioimaging” was interesting and well written by the authors. All the necessary characterizations were done and the discussion has been supported by the experimental data.
Response 1: Thank you for your careful review and positive evaluation of our work.
Point 2: However, the manuscript was very confusing because the authors presented some properties of BPN@NaGdF4 and some properties of BPN@NaYF4 nanocomposite. For example, photoacoustic property for BPN@NaGdF4 nanocomposite and luminescence property for BPN@NaYF4 nanocomposite. It would be better if the authors present the comparison between the photoacoustic and luminescence properties of these two nanocomposites.
Response 2: Thank you for your comments. This work presents a simple method to prepare the nanocomosites containing BPN and rare earth nanoparticles. The NaGdF4 and NaYF4 are the most typical lathanide-doped nanoparticles. The value of the synthesis of BPN@NaReF4 nanocomposites lies in the applications of bimodal bioimaging. For example, integration of BPN with NaGdF4 can realize photoacoustic and magnetic resonance imaging, while the BPN@NaYF4: Yb, Er have the potential photoacoustic and fluorescence bioimaging. In this case, we just list the photoacoustic property of BPN, magnetic resonance imaging of BPN@NaGdF4 and fluorescence property of BPN@NaYF4: Yb, Er.
Point 3: Also, the authors need to present SEM images of the BPN nanosheet, NaGdF4 nanoparticles, and the BPN@NaGdF4 nanocomposite.
Response 3: Thank you for your suggestion. We got the SEM images of BPN, NaGdF4 nanoparticles, and the BPN@NaGdF4 nanocomposite in Figure S5. Because of the poor conductivity of BPN and NaGdF4, SEM can not capture clear images.

Reviewer 2 Report
After going through the paper, this black phosphor work is interesting and can be accepted in its current form, recommend it for publication.
Author Response
We are extremely grateful to reviewer for the positive evaluation of our work.
Reviewer 3 Report
Line 12, the author should define the BPN after where they mentioned Black phosphorus nanomaterials.
Line 80, the title of the section should be shifted to the next page.
A careful proof read should be done by authors.
The authors need to expand on the reference lists. For example the work of Harry Dorn from the Chemistry Department of Virginia Tech should be cited as he is one of the pioneers of using Rare Earths for MRI.
Also this paper could be relevant to be cited: Phys. Rev. B 56, 4518 – Published 15 August 1997
Author Response
Point 1: Line 12, the author should define the BPN after where they mentioned Black phosphorus nanomaterials.
Line 80, the title of the section should be shifted to the next page.
A careful proof read should be done by authors.
Response 1: Thank you for your comments. We have revised the manuscript. Any changes are highlighted in red.
Point 2: The authors need to expand on the reference lists. For example the work of Harry Dorn from the Chemistry Department of Virginia Tech should be cited as he is one of the pioneers of using Rare Earths for MRI.
Also this paper could be relevant to be cited: Phys. Rev. B 56, 4518 – Published 15 August 1997
Response 2: Thank you for your suggestions of reference. The work of Harry Dorn was cited as ref [14]. The reference “Phys. Rev. B 56, 4518” was cited as ref [13].
Round 2
Reviewer 1 Report
The authors responded to all the queries raised by me and the manuscript was ready for publication in its present form.